# Electron-phonon coupling and vibrational properties of size-selected linear carbon chains by resonance Raman scattering

P. Marabotti [1], M. Tommasini [2], C. Castiglioni [2], P. Serafini[1], S. Peggiani[1], M. Tortora[3], B. Rossi[3], A. Li Bassi [1], V. Russo[1] & C. S. Casari [1] ✉

UV resonance Raman spectroscopy of size-selected linear sp-carbon chains unveils vibrational overtones and combinations up to the fifth order. Thanks to the tunability of the synchrotron source, we excited each H-terminated poly-yne (HC$_n$H with $n$ = 8,10,12) to the maxima of its vibronic absorption spectrum allowing us to precisely determine the electronic and vibrational structure of the ground and excited states for the main observed vibrational mode. Selected transitions are shown to enhance specific overtone orders in the Raman spectrum in a specific way that can be explained by a simple analytical model based on Albrecht's theory of resonance Raman scattering. The determined Huang–Rhys factors indicate a strong and size-dependent electron-phonon coupling increasing with the sp-carbon chain length.

Besides the most known carbon allotropes, mainly based on sp$^2$ hybridization, linear sp-carbon chains have recently attracted the interest of the scientific community as carbon atomic wires modeling the elusive carbon allotrope carbyne and showing appealing mechanical, thermal, and optoelectronic properties[1–5]. Finite sp-carbon wires can be synthesized up to a few tens of carbon atoms and show size-dependent optical and electronic properties[6,7]. Long carbon wires exceeding thousands of atoms encapsulated in carbon nanotubes (i.e., confined carbyne) have provided a way to approach the model carbyne whose properties should no longer depend on the length of the system[1,8,9]. Recently, the optical gap behavior in finite systems (i.e., oligoynes) as a function of chain length has been shown to reach saturation, thus pointing to the possible value of the ideal carbyne[8,10]. However, the question of how the properties change from short carbon wires to carbyne remains largely unanswered.

Raman spectroscopy has proven to be a fundamental technique for the characterization of materials. The Raman process, based on the inelastic scattering of photons by phonons or molecular vibrations, can be enhanced in resonance Raman scattering by using photon energies matching the absorption edges of the material and thus involving excited electronic states. Resonance Raman scattering can easily detect higher-order scattering processes involving more than one phonon or vibrational quantum. For instance, resonance Raman scattering and electron-phonon coupling are relevant in understanding the spectrum of graphite, single-walled carbon nanotubes, and graphene[11–17]. The so-called 2D mode, sensitive to the number of graphene layers, is a second-order Raman mode (i.e., the overtone of the D peak) showing a larger intensity than the first order only in single layer graphene[12,17]. While Raman excitation profiles of the fundamental Raman transition have been extensively investigated in the past for many systems including carbon nanotubes and conjugated molecules[18–22], the effect of the resonance with different vibronic states in the region of the overtones has not been observed so far, to the best of our knowledge.

Sp-carbon chains feature a specific Raman fingerprint in a spectral region (i.e., 1800–2300 cm$^{-1}$) where other carbon systems do not present any signal. The main vibrational mode, also called α or C-mode, involves a collective motion of all the CC bonds of the chain, as discussed in the framework of the Effective Conjugation Coordinate ECC model, and carries fundamental information about the structure and properties of sp-carbon chains[3,23–27]. Sp-carbon features a larger Raman cross section than sp$^2$ amorphous carbon and confined carbyne has been recently identified as the strongest Raman scatterer ever

[1]Micro and Nanostructured Materials Laboratory—NanoLab, Department of Energy, Politecnico di Milano via Ponzio 34/3, I-20133 Milano, Italy. [2]Department of Chemistry, Materials and Chem. Eng. 'G. Natta', Politecnico di Milano Piazza Leonardo da Vinci 32, I-20133 Milano, Italy. [3]Elettra Sincrotrone Trieste, S.S. 114 km 163.5, Basovizza, 34149 Trieste, Italy. ✉e-mail: carlo.casari@polimi.it

reported[28,29]. The resonance Raman cross section exceeding any other material by two orders of magnitude makes confined carbyne a candidate for nanoscale temperature monitoring by exploiting the Stokes/anti-Stokes ratio[30].

Resonance Raman spectroscopy has been rarely employed to investigate the vibrational spectra of short sp-carbon systems[31–34]. Pre-resonance and resonance conditions have been exploited for polyynes absorbing in the UV[31,35]. Recently resonance Raman in the visible range has been used to infer the excitation profile of confined carbynes[9]. The evaluation of an average Huang–Rhys factor of 1.82 indicates a large electron-phonon coupling and accounts for the extremely high Raman activity of this system[8,9]. In this framework, the size-dependent resonance Raman behavior and the electron-phonon coupling in the limit of short sp-carbon wires is an open issue.

In this work, we provide a resonance Raman investigation of size-selected H-terminated polyynes, as the simplest model of sp-carbon wires in the short size limit. Such systems represent the simplest linear carbon chain comprising sp-hybridized carbon atoms only and prove to be the ideal system to investigate resonance Raman processes in detail. We exploited synchrotron radiation as a tunable coherent UV light source to precisely excite the size-dependent absorption vibronic maxima resolving by resonance Raman scattering the vibrational fine structure (up to the fifth level) of the ground state. We evaluated the electron-phonon coupling (Huang–Rhys factor) and the vibrational structure of the excited state through the analysis of the UV-Vis absorption spectra. The Raman spectrum up to the fifth-order shows a peculiar behavior of the overtone intensity as a function of the excitation at selected vibronic transitions that can be explained by Albrecht's theory of resonance Raman scattering in a simple and analytic approach. This method allows also identifying the single

contribution of the two main Raman active modes (α and β[25]) in the Huang–Rhys factor of short polyynes.

## Results

Resonance Raman spectra of $C_8$ have been acquired at three excitation wavelengths, i.e., 226, 216, and 206 nm (see Fig. 1). These wavelengths match the resonance condition with individual vibronic transitions, as evidenced by the UV-vis absorption spectrum in Fig. 1a, and correspond to $|0\rangle_g \rightarrow |0\rangle_e$, $|0\rangle_g \rightarrow |1\rangle_e$, and $|0\rangle_g \rightarrow |2\rangle_e$, where $|0\rangle_g$ is the first vibrational level of the ground state, and $|k\rangle_e$ is the $k^{th}$ vibrational level of the excited state (see Supplementary Fig. 1b and discussion below). It is reasonable to assume the lowest vibrational level as the initial state of the Raman transitions ($|i\rangle_g = |0\rangle_g$) since the α modes of polyynes ($v \cong 1800$–$2200 \, cm^{-1}$, with vibrational quantum energy of about 0.22–0.27 eV) have a negligible population of the higher vibrational levels given the available ambient thermal energy at 288 K (0.0248 eV). In all the reported resonance Raman spectra, the vibrational features of the solvent (mixture of acetonitrile and water) are present in the first-order Raman region up to 3700 $cm^{-1}$ (see Fig. 1). We adopted the CN stretching band of acetonitrile at 2258 $cm^{-1}$ as an internal reference across the whole dataset. The other solvent features are the combination band of acetonitrile at 2290 $cm^{-1}$, the symmetric and anti-symmetric CH stretching of acetonitrile at 2940 $cm^{-1}$ and 3000 $cm^{-1}$, respectively, and the broad OH stretching band of water extending from 3100 to 3700 $cm^{-1}$ [36,37].

Independently on the excitation wavelength, the characteristic α line of $C_8$ is stably located nearby 2172 $cm^{-1}$, very close to the CN stretching mode of acetonitrile. The position and the shape of this band correspond to data reported by ref. 25 in off-resonance conditions (2172 $cm^{-1}$). The fundamental α line is as intense as the solvent

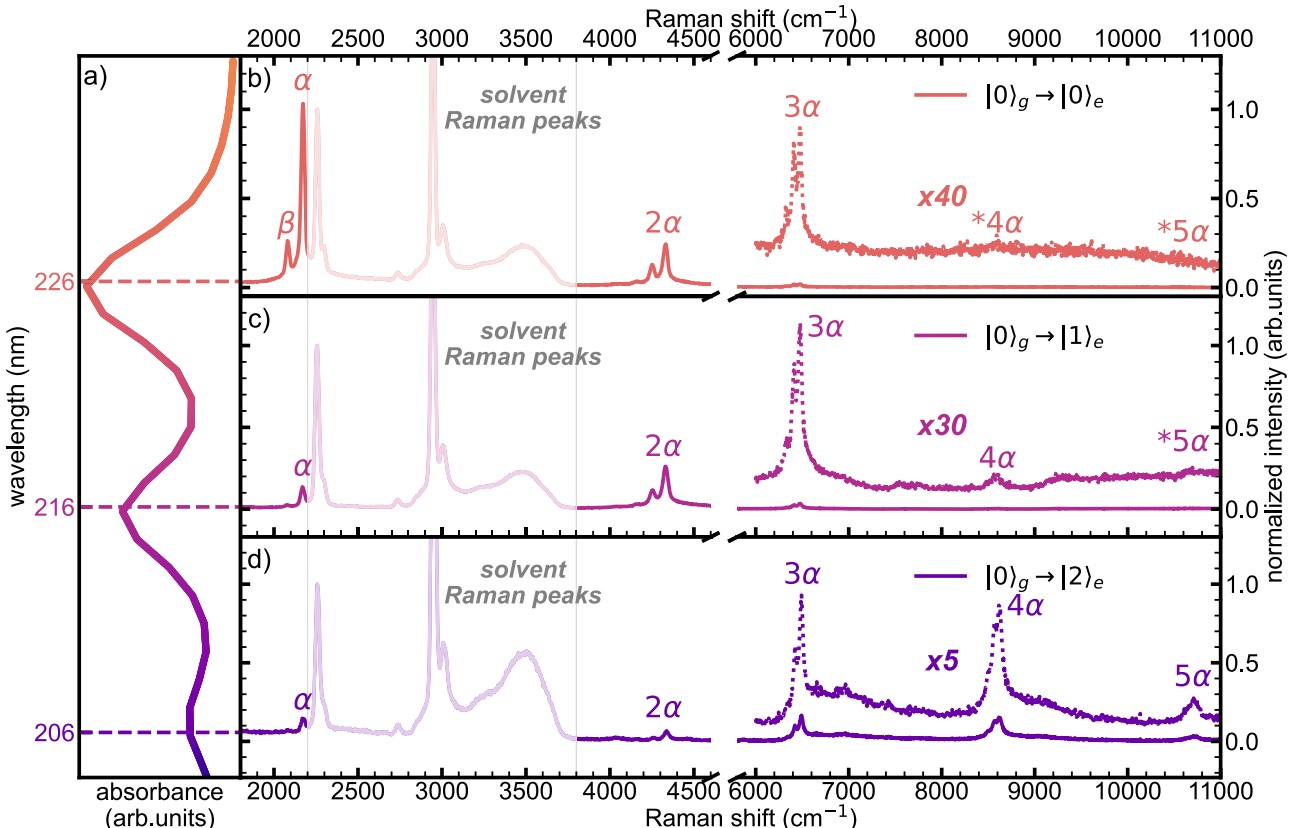

**Fig. 1 | UV-Vis absorption spectrum and UV resonance Raman spectra of $C_8$.** UV-Vis absorption spectrum (panel **a**) and UV resonance Raman spectra of $C_8$, excited using synchrotron radiation at three different wavelengths: 226 nm ($|0\rangle_g \rightarrow |0\rangle_e$, **b**), 216 nm ($|0\rangle_g \rightarrow |1\rangle_e$, **c**), and 206 nm ($|0\rangle_g \rightarrow |2\rangle_e$, **d**). The signals above 6000 $cm^{-1}$ in all panels are magnified to highlight low-intensity signals. The vibrational lines of the solvent are shadowed by white boxes. The spectra are normalized to the CN stretching peaks of acetonitrile.

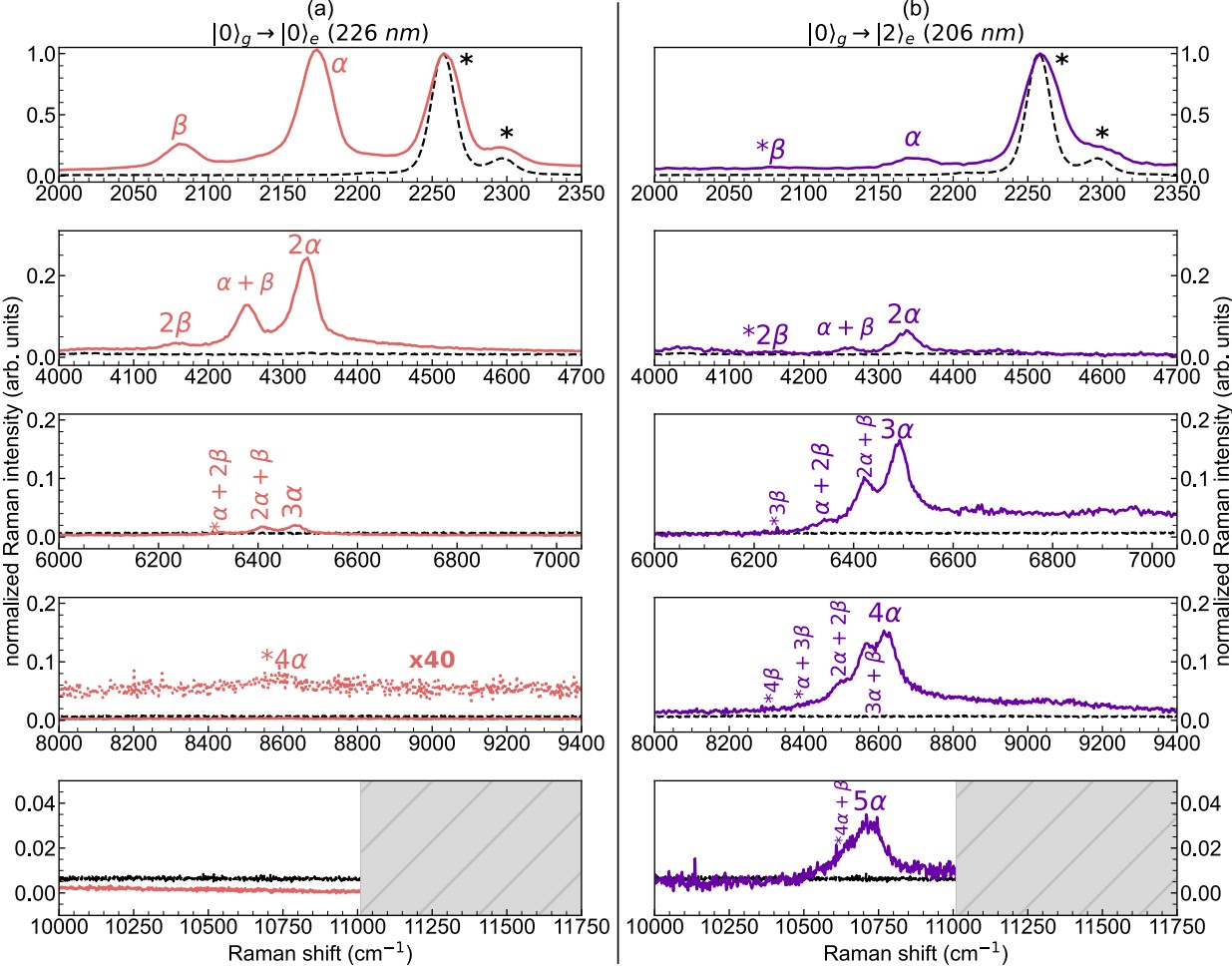

**Fig. 2 | Comparison between UV resonance Raman spectra of C$_8$ at different resonance conditions.** Comparison between vibrational bands of C$_8$ excited at two different resonances, namely 226 nm ($|0\rangle_g \to |0\rangle_e$, **a**) and 206 nm ($|0\rangle_g \to |2\rangle_e$, **b**). First- and high-orders α and β peaks are shown, as well as combination bands peaks ($m\alpha + n\beta$). The black dashed lines in all the panels are the spectra of the solvent, while the specific bands of acetonitrile are highlighted by asterisks in the plots in the first row. The spectra are normalized to the CN stretching band of acetonitrile. The expected positions of some overtones and combinations, too weak to be reliably detected, are identified with a star (*) prefix.

bands when exciting in resonance with the first ($k = 0$) vibronic transition (226 nm), despite the low concentration of C$_8$ (see Supplementary Table 1), while it is much less intense when tuning the synchrotron radiation to 216 nm ($k = 1$) and 206 nm ($k = 2$) (see Figs. 1 and 2). Moreover, at 226 nm excitation, we observe a satellite peak close to the α line at 2082 cm$^{-1}$ (named β line[25]), which is assigned to another CC stretching mode of the sp carbon chain[3,23,25]. The intensity of the β line is always lower than that of the α line, being almost non-detectable at 206 nm ($k = 2$).

In the region beyond the OH stretching of water (>4000 cm$^{-1}$), resonance Raman spectra of C$_8$ at different excitation energies show features located at approximately two, three, four, and five times the Raman shift of the α and β peaks, immediately ascribed to overtones ($m\alpha$ or $n\beta$, with $m$ and $n$ being the number of vibrational quanta) and combination bands ($m\alpha + n\beta$)—see Figs. 1 and 2. In particular, the second-order region (4000–4400 cm$^{-1}$) shows 3 peaks that correspond to the first overtones and combination of the α and β lines, namely 2α, 2β, and α+β. The three quanta region (6000–6600 cm$^{-1}$) allows identifying the expected four overtones and combinations (3α, 2α+β, α+2β, 3β). The fourth-order Raman region (8000–8800 cm$^{-1}$) shows four of its five distinctive overtones and combinations (4α, 3α +β, 2α+2β, α+3β, 4β), only when in resonance with the $|0\rangle_g \to |2\rangle_e$

vibronic transition at 206 nm. Indeed, only weak features, hardly detectable in the background, can be seen when in resonance with the $|0\rangle_g \to |0\rangle_e$ vibronic transition at 226 nm. Such difference is more evident for the fifth-order region where a rather broad band centered at about 10,700 cm$^{-1}$ is detectable only when exciting at 206 nm ($k = 2$) and represents the convolution of the six expected overtones and combinations. In the measured UVRR spectra, we also observe that combination modes with increasing β contribution show decreasing intensities within the same overtone manifold. This is expected because the Raman intensity of the β line is significantly smaller than the intensity of the α line, and the Raman intensity of combinations can be approximated—in resonance condition—by the product of the intensities of the related fundamental Raman transitions[38].

## Discussion

The combined use of the experimental UV-Vis absorption and resonance Raman spectra allowed us to investigate the vibrational fine structure of the α mode in short polyynes. From the resonance Raman spectra recorded at different excitation wavelengths, we have determined the vibrational levels of the ground state of the α mode up to $|5\rangle_g$, as shown in Fig. 3 for C$_8$. The position of each vibrational level is the energy of the fundamental transition of the α mode and its

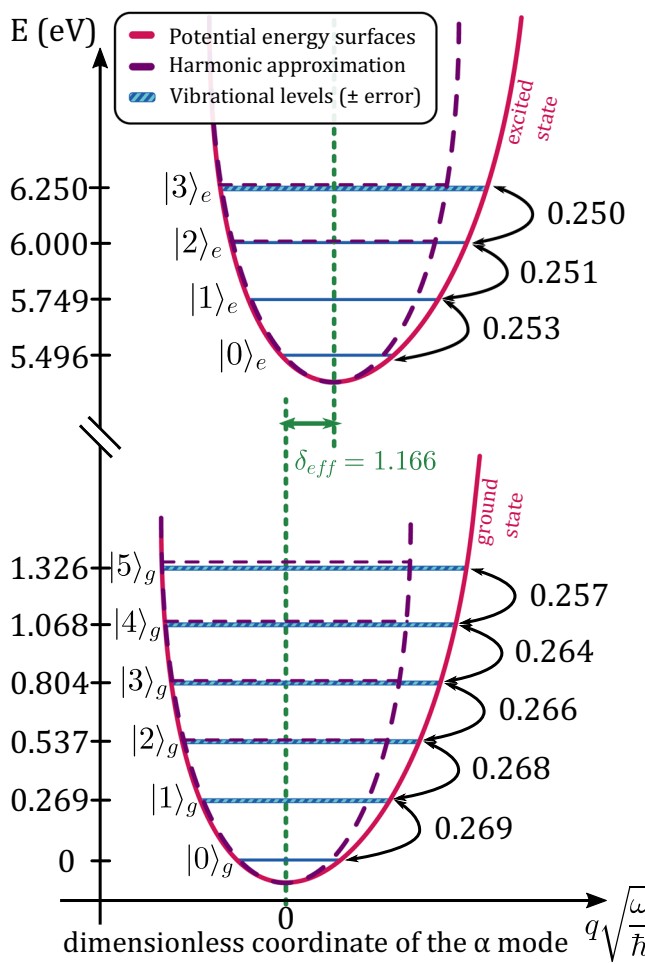

**Fig. 3 | Vibrational energy diagrams of ground and excited states of the α mode of C₈.** Vibrational diagram of the ground and excited potential energy surfaces extracted from experimental UVRR and UV-Vis absorption spectra of C₈ and referred to the normal coordinate $q$ of the α mode. The energy (in eV) of each vibrational level $|k\rangle_j$ of the $j^{th}$ state is reported in the left axis, while the energy of the $|0\rangle_g$ level is fixed to 0 eV. The thickness of the vibronic lines represents the error of the fit model used to deconvolve the UVRR and UV-Vis absorption spectra. The spacing between neighboring vibrational levels ($\Omega_{k,j}$) is displayed on the right in eV and indicated by black arrows. The purple dashed curves and lines represent the harmonic approximation of the potential energy surfaces and vibrational levels, respectively. The effective displacement parameter $\delta_{eff}$ is reported in green.

overtones. For the excited state, the energies of the vibrational levels cannot be obtained directly from the UV-Vis absorption spectrum since it is not possible to identify the contribution of the sole α transition in the broad peaks of the vibronic progression, which of course also implies the β mode, and possibly other totally symmetric vibrations. However, since the Raman intensity of the other modes (e.g., the β mode) is much smaller than the α mode, we adopt here a one-mode approximation, which is also *a posteriori* justified by the analysis of the Franck–Condon integrals (Huang–Rhys factors) reported in the Supplementary Discussion. Thus, we can roughly estimate the energies of the vibrational levels of the excited state of the α mode up to $|3\rangle_e$, as shown in Fig. 3 for C₈. Similar results are obtained for C₁₀ and C₁₂ (see Supplementary Fig. 4).

The anharmonicity of the potential energy surface of the α mode can be determined by comparing the spacing between the vibrational levels (see arrows in Fig. 3). Considering both the ground and the excited states of C₈, the energy spacing extracted from experimental

data decreases by increasing the vibrational level. However, this lowering is nearly negligible compared to the harmonic spacing given by the $|0\rangle_g \rightarrow |1\rangle_g$ energy difference, and reaches a maximum of 2% and 1.4% in the case of the ground and excited states, respectively. Hence, in the following, we will assume the harmonic approximation for both the ground and excited state potential energy surfaces. Moreover, the spacing between experimentally determined vibrational levels is almost equal in the ground and excited states, e.g., 0.269 eV and 0.253 eV, respectively, for C₈ (see Fig. 3). Thus, it is reasonable to adopt the same harmonic approximation of the potential energy surfaces for the ground and excited states. Hence, by the analysis of the clear vibronic progression of polyynes, and by the observation of high order Raman features, made possible by the high enhancement and resolution of the SR-based UVRR setup at Elettra, we can precisely assess the vibronic levels of polyynes despite the very low concentration of the samples.

In this framework, an important parameter to evaluate the optoelectronic properties of conjugated systems is the strength of the electron-phonon coupling. We can investigate it by calculating the Huang–Rhys (HR) factor, $S$, of polyynes. Indeed, in the Franck–Condon model, this nondimensional parameter expresses the average number of quanta involved in the vibrational transition. By keeping the one-mode approximation introduced before, it is possible to quantitatively evaluate the HR factor from the UV-Vis spectrum[39]. Thus, assuming we neglect the effect of higher vibrational levels as initial states in the transition (as discussed above), we can apply the following relation[39–41]

$$\frac{I_{0\rightarrow\nu}}{I_{0\rightarrow0}} = \frac{S^{\nu}}{\nu!} \tag{1}$$

which shows that the $\nu$-th power of the HR factor $S$ is proportional to the ratio of the intensity of the $\nu$-th vibronic transition compared to the fundamental transition. From Eq. (1), we obtain the HR factors for the $\nu = 1$ transition of all the H-capped polyynes we have discovered so far, i.e., from C₆ to C₂₆, whose UV-Vis spectra are reported in Supplementary Fig. 5. The standard deviations reported in Fig. 4 are calculated from the fit errors of the UV-Vis spectra of polyynes used to extract the intensity of each vibronic peak.

Within this one-mode model, we can estimate a dimensionless effective displacement parameter ($\delta_{eff}$), proportional to the distance between the equilibrium position of the ground and excited states. Indeed, the displacement parameter determines both the intensities of the vibronic progressions in UV-Vis absorption spectra and the activities of the collective vibrational modes in Raman spectra of polyynes[25,42]. Moreover, it is related to the thermal stability, optoelectronic and quantum chemical properties of molecules. It turns out that the effective displacement parameter is connected to the HR factor of the $\nu = 1$ transition (i.e., $S = \delta_{eff}^2/2$—see Supplementary Discussion)[43,44]. In this way, we can directly evaluate the effective displacement parameter (i.e., $\delta_{eff} = \sqrt{2S}$), from the HR factors of all H-capped polyynes, as reported in Fig. 4b. In particular, we found $\delta_{eff} = 1.166 \pm 0.04$ for C₈ (see also Fig. 3), $1.183 \pm 0.06$ for C₁₀ and $1.225 \pm 0.04$ for C₁₂ (see also Supplementary Fig. 4). Both the HR factors and the effective displacement parameters $\delta_{eff}$ grow as the length of H-capped polyynes increases, showing a strong correlation between the electron-phonon coupling and π-conjugation in linear sp-carbon chains. The values of the HR factor of H-capped polyynes reported in Fig. 4a are larger than those of the radial breathing mode of carbon nanotubes[11,20], and are comparable with the HR factors found for β-carotene[21,22,41]. Recently, Martinati et al. calculated the HR factor of long sp-carbon chains (≫ 100 carbon atoms) confined in carbon nanotubes from wavelength-dependent resonance Raman measurements and found an average value of 1.82 as a possible upper limit for long chains towards carbyne[9]. Our results confirm the increasing trend of the HR factor with increasing chain length and π-conjugation.

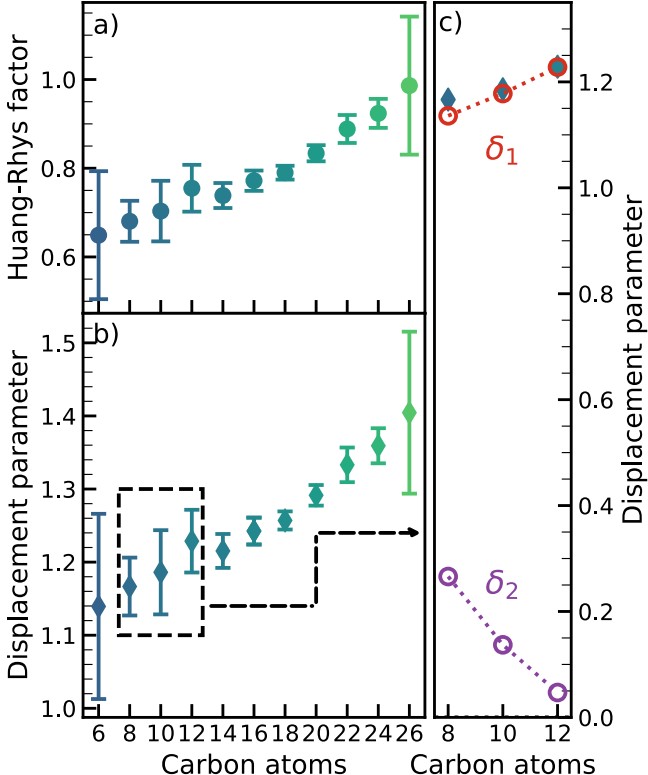

**Fig. 4 | Huang–Rhys and effective displacement parameters of hydrogen-capped polyynes. a** Huang–Rhys factors $S$ and **b** effective nondimensional displacement parameter $\delta_{eff}$ of H-capped polyynes with increasing length extracted from UV-Vis absorption spectra (see Supplementary Fig. 5). **c** Effective nondimensional displacement parameter $\delta_{eff}$ (blueish diamonds) calculated from HR factors (see Eq. (1)) extracted from experimental UV-Vis absorption spectra of $C_8$, $C_{10}$, and $C_{12}$. The displacement parameters for the α ($\delta_1$, red dots) and β ($\delta_2$, purple dots) modes of $C_8$, $C_{10}$, and $C_{12}$, calculated from UVRR and UV-Vis absorption spectra with the method explained in Supplementary Discussion, are also reported for comparison. The errorbars of panels **a** and **b** represents the error of the fit model used to deconvolve the UV-Vis absorption spectra.

In discussing the Huang–Rhys factor, the analysis of the UV-Vis spectra was limited to a one-mode picture. However, polyynes feature other modes besides the α mode in their Raman spectra. Indeed, as can be seen in both Figs. 1 and 2 for $C_8$, we can easily detect the β mode, its overtones, and combinations with the α mode. The intensity of the overtones and combination modes follow an intriguing behavior as a function of the excitation energy. By exciting the $|0\rangle_g \rightarrow |0\rangle_e$ transition of $C_8$ at 226 nm, the intensity of the α mode monotonically decreases by increasing the order of the Raman mode. However, such a trend completely changes by exciting $C_8$ at 216 nm and 206 nm. This is testified by the relative intensities of the fourth- and fifth-order Raman peaks that are detectable by exciting at 206 nm (see Fig. 2b), whereas they are extremely weak or undetected with the excitations at 226 nm and 216 nm. Such remarkable intensity behavior indicates the occurrence of selective intensification of multiple quanta Raman transitions, showing a cross section that in several cases exceeds that of the fundamental mode. This phenomenon is a peculiar feature of the resonance Raman effect, observed here in polyynes for the first time, thanks to the availability of high-order transitions in the UVRR spectra. We observe similar results in the spectra of $C_{10}$ and $C_{12}$ (see Supplementary Figs. 2 and 3), where we detected up to the fourth-order Raman peaks of their α modes. We also notice that the β line weakens by increasing the polyyne length, and, as a consequence, the overtone/combination regions have a less structured appearance than in the case of $C_8$.

In this framework, a two-mode (α and β) model is required to explain the experimental results, based on Albrecht's theory of resonance Raman[45] (see Supplementary Discussion) and worked out in the hypothesis of resonance with specific vibronic transitions. The expressions of Raman intensities so obtained require the evaluation of the displacement parameters related to the α and β modes, $\delta_1$ and $\delta_2$, respectively. In this way, we can predict the intensity pattern of the α mode and its overtones in the UVRR spectra of polyynes. As illustrated in the Supplementary Discussion, we have determined $\delta_1$ and $\delta_2$ from experimental data by combining the UV-Vis absorption and first-order resonance Raman spectra of each polyyne (see Supplementary Discussion), which resulted in the values of 1.136 and 0.266 for $C_8$, 1.178, and 0.137 for $C_{10}$, and 1.228 and 0.047 for $C_{12}$, respectively. Interestingly, the values of $\delta_1$ (α mode) approach those of $\delta_{eff}$, while $\delta_2$ (β mode) goes to zero as the chain length increases, as shown in Fig. 4c. Indeed, the effective displacement parameter is connected to $\delta_1$ and $\delta_2$ according to the relationship $S = \frac{\delta_{eff}^2}{2} = \frac{\delta_1^2}{2} + \frac{\delta_2^2}{2}$ (see Supplementary Discussion[43,44]). This occurrence makes the one-mode approximation increasingly more reasonable as the length of the chain increases and it is consistent with the expected decrease in the Raman activity of the β mode which is ultimately not present in the infinite chain[1,3]. Based on this observation, we expect that the fine details in the resonance Raman trends are harder to describe with the one-mode approximation in $C_8$ than in $C_{12}$ and longer chains, whose $\delta_2$ is expected to approach zero.

The experimental behavior of the relative Raman intensities of the $m$-th overtones of the α mode ($I_{0\rightarrow m}/I_{0\rightarrow 1}$) in resonance with different vibronic transitions ($|0\rangle_g \rightarrow |k\rangle_e$) is reported in Fig. 5. The intensity of each $m\alpha$ peak has been normalized to that of the fundamental ($m = 1$) transition. By exciting $C_8$ at 226 nm ($|0\rangle_g \rightarrow |0\rangle_e$), the intensity ratio follows a decreasing progression vs. the $m$ quantum number of the overtone, as expected in non-resonance Raman scattering[46]. However, by exciting $C_8$ at 216 nm ($|0\rangle_g \rightarrow |1\rangle_e$), the intensity of the second-order ($2\alpha$) line exceeds that of the first order. Remarkably, at 206 nm ($|0\rangle_g \rightarrow |2\rangle_e$), the relative intensity maximum is reached with the fourth overtone ($4\alpha$). We also observe a strong modulation of the relative intensities of the $m\alpha$ lines for $C_{10}$ and $C_{12}$, as reported in Fig. 5b, c, respectively. By applying Albrecht formalism, we can compute the values of the ratio of the intensity of the $m$-th overtone to the first-order mode by the following expression (see Supplementary Discussion):

$$R_{km} = \frac{I_{0\rightarrow m}}{I_{0\rightarrow 1}} \approx \left[\frac{1 - m\epsilon}{1 - \epsilon}\right]^4 \left[\frac{e\langle k|m\rangle_g}{e\langle k|1\rangle_g}\right]^2 \qquad (2)$$

where $k$ identifies the specific vibronic resonance ($|0\rangle_g \rightarrow |k\rangle_e$) and $m = 2, 3, \ldots$ labels the first, second, ... Raman overtone. In Eq. (2) we set $\epsilon = \omega_\alpha/\omega_0$, where $\hbar\omega_\alpha$ and $\hbar\omega_0$ are the quantum energies of the α mode and the UV excitation, respectively (for instance, for $C_8$ at the 226 nm excitation, it is $\epsilon = 0.049$).

In deriving Eq. (2) (see Supplementary Discussion), we have assumed the same harmonic approximation of the potential energy surfaces for the ground and excited states; this choice is justified by the previous analysis of the vibrational structure of the two electronic states. The displacement parameter $\delta_1$ rules the overlap integrals between the vibrational wavefunctions of the ground and excited state (Franck–Condon factors) which appear in Eq. (2), as illustrated in Supplementary Discussion. The relative intensities of the overtones of the α line, obtained by Eq. (2), are reported in Fig. 5. By comparing experimental and theoretical data of Fig. 5, we observe that the model can capture most of the observed modulation of the relative intensities of the overtones for the different resonance conditions, in particular for the $|0\rangle_g \rightarrow |0\rangle_e$ ($C_8$ and $C_{10}$) and the $|0\rangle_g \rightarrow |1\rangle_e$ ($C_8$, $C_{10}$, and $C_{12}$)

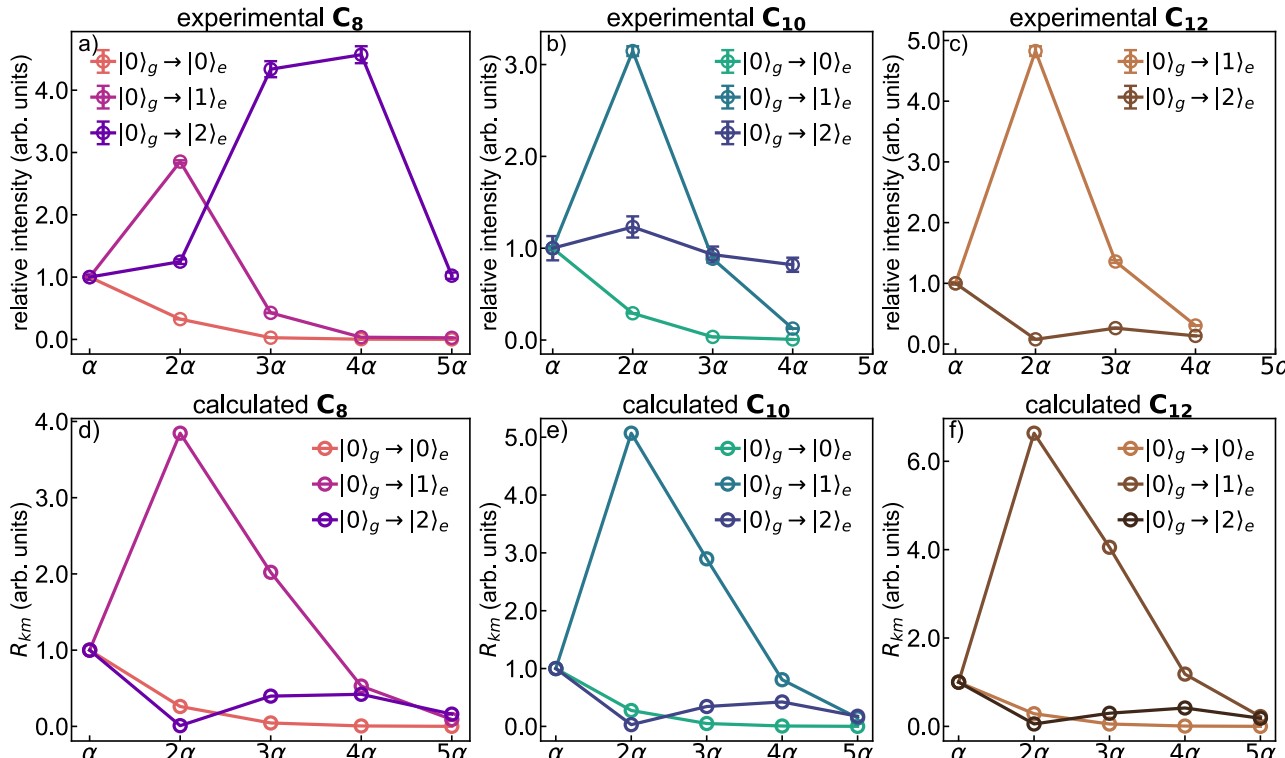

**Fig. 5 | Experimental and calculated relative intensities of the overtones of the α mode of C$_8$, C$_{10}$, and C$_{12}$.** Relative intensities of the overtones of the α mode of C$_8$ (**a**), C$_{10}$ (**b**), C$_{12}$ (**c**) determined as peak intensities compared to that of the first-order α mode of the experimental resonance Raman spectra excited at different wavelengths, corresponding with the resonance conditions listed in Supplementary Table 1. The spectrum corresponding to the absorption maximum of C$_{12}$ (273 nm, $|0\rangle_g \rightarrow |0\rangle_e$) could not be recorded due to limitations in the undulator gap aperture. The errorbars of panels **a**–**c** represents the error of the fit model used to deconvolve the UVRR spectra. Panels from **d**–**f** display the prediction of the relative intensity of the *m*-th overtone of the Raman α line of C$_8$ (**a**), C$_{10}$ (**b**), and C$_{12}$ (**c**) as a function of the resonance condition, as computed according to Eq. (2).

vibronic transitions. However, in the case of the $|0\rangle_g \rightarrow |2\rangle_e$ resonance condition, the model does not fully account for the observed different behaviors of the three polyynes, and the predicted trends are fairly similar from C$_8$ to C$_{12}$. This is expected because the adopted assumptions can lose their robustness for Raman transitions involving higher *k*. The analysis of the terms appearing in Eq. (2), allows the rationalization of the observed intensity trends.

Since in Eq. (2) the prefactor containing $\epsilon$ decreases monotonically with *m*, the most relevant term that can account for the non-monotonic behaviors experimentally observed (Fig. 5) depends on the Franck–Condon integrals[18]. For systems with vibronic progressions dominated by the 0–0 and 0–1 transitions, which is typical of systems with a $\delta$ value close to 1, the overlap between the *k*–th and the *m*–th vibrational wavefunction of the electronic excited (*e*) and ground (*g*) state in $_e\langle k|m\rangle_g$ is usually significant when $k = m$ and/or $k = m \pm 1$. This explains the remarkable intensity of high-order overtones observed when the experiment is done in resonance with vibronic states characterized by *k* values larger than zero.

Based on these results, polyynes display ideal features that allow investigating their vibrational and electronic properties by resonance Raman. First, the Raman modes in the high-frequency region result in well-separated vibronic peaks in the absorption spectra. This allows precisely selecting the transition by matching the excitation wavelength. Second, simple analytic expressions can be used to describe the resonance process thanks to the presence of just two relevant Raman active modes. In general, the rapid vanishing of the β mode intensity with increasing polyyne length validates the use of an even simpler one-mode description. Such features emerge from the properties of an sp-carbon backbone with the simplest possible terminations, i.e., by one hydrogen atom at each side, which avoids additional resonance-enhanced Raman modes and possibly a more complex

absorption profile induced by the presence of end groups with π electrons (e.g., phenyl groups), often showing conjugation with the π orbitals of the sp chain. Third, the huge resonance Raman cross section of polyynes allows easy detection even at very low sample concentrations. Remarkably, the model and the strategy adopted to justify the intensity pattern exhibited by the overtones in the resonance Raman of sp-carbon chains are valid for polyynes of any length. For longer chains, one expects that the α mode is dominant since $\delta_2$ becomes negligible already for C$_{12}$. In such conditions, it can be shown that the dispersion of the α mode as a function of the chain length monotonically approaches the limit of about 1900 cm$^{-1}$ for an isolated chain of infinite length[47]. Furthermore, we expect to observe similar intensity modulations at different resonance conditions as those studied in this work. Another effect worth investigating with resonance Raman spectroscopy is the behavior of polyynes in different solvents and/or encapsulated in carbon nanotubes. We may expect a similar qualitative behavior as that discussed here for C$_8$, C$_{10}$, and C$_{12}$, with a possible modulation of the $\delta$ parameter because of the interaction effects with the environment. For instance, such effects have been highlighted by comparing the Raman spectra of hydrogen-capped polyynes in solution with those of the same polyynes encapsulated in carbon nanotubes which shows a remarkable decrease of the position of the α mode by a few tens of wavenumbers[48]. Similar experiments have been performed on long linear carbon chains or confined carbyne in carbon nanotubes showing the effects of the confinement on the Raman bands of sp-carbon chains[49–54].

In summary, by exploiting the fine wavelength tunability of the synchrotron radiation, we accomplished the first experimental detection of high-order (up to the IV/V) overtones of the α mode of polyynes, including the combination bands with another collective CC stretching mode (β mode). Based on our results, the detailed

electronic and vibrational structure of the ground and first excited state in polyynes can be derived with a one-mode approximation dominated by the α mode. The values of the Huang–Rhys factors determined in such linear carbon structures characterized by π-conjugated electrons confined at the nano and molecular scale show a strong and size-dependent electron-phonon coupling, which is appealing for potential exploitation in optoelectronic applications.

We modeled the resonance effect in the framework of Albrecht's theory of resonance Raman which allowed us to justify the relative intensities of the Raman transitions detected at high vibrational quanta. Polyynes prove to be an ideal system for multi-wavelength resonance Raman spectroscopy, which allows the precise investigation of high-order vibrational transitions due to their well-resolved vibronic features and high-frequency Raman modes.

## Methods

### Synthesis and separation of hydrogen-capped polyynes
Solutions of size-selected ($n = 8,10,12$) hydrogen-capped polyynes (HC$_n$H, from now on C$_n$) were synthesized by pulsed laser ablation in acetonitrile, detected, and separated by high-performance liquid chromatography employing the experimental method described in ref. 55. Their UV-Vis spectra and concentrations are reported in Supplementary Fig. 1a and Supplementary Table 1, respectively.

### UV resonance Raman spectroscopy measurements
UV resonance Raman (UVRR) spectra were collected by exploiting the synchrotron-based Raman set-up available at the BL10.2-IUVS beamline of Elettra Sincrotrone Trieste (Italy)[56]. All the samples were measured at a fixed temperature of 288 K using different excitation wavelengths in the deep UV range (see Supplementary Table 1), provided by the emission of synchrotron radiation (SR) source. The exciting wavelengths were set by regulating the undulator gap aperture and using a Czerny-Turner monochromator (Acton SP2750, focal length 750 mm, Princeton Instruments, Acton, MA, USA) equipped with holographic gratings with 3600 groves/mm for monochromatizing the incoming SR. Raman signal was collected in backscattered geometry, by a single pass of a Czerny-Turner spectrometer (Trivista 557, Princeton Instruments, 750 mm of focal length) equipped with a holographic grating at 1800g/mm and 3600 g/mm. The spectral resolution was set at $\frac{spectral\ range}{1340}$ cm$^{-1}$/pixel, depending on the excitation wavelength (e.g., 2.9 cm$^{-1}$/pixel at 206 nm and 1.7 cm$^{-1}$/pixel at 261 nm). The calibration of the spectrometer was standardized using cyclohexane (spectroscopic grade, Sigma Aldrich). The final radiation power on the samples was kept between a few up to tens of μW (see Supplementary Table 1). Any possible photo-damage effect due to prolonged exposure of the sample to UV radiation was avoided by continuously spinning the sample cell during the measurements.

### Fit of UV-Vis absorption and UV resonance Raman spectra
The calculation of the Huang–Rhys factors and nondimensional displacement parameters requires a joint analysis of the UV-Vis absorption and UV resonance Raman spectra (see Supplementary Discussion). Concerning UV-Vis absorption spectra of C$_8$, C$_{10}$, and C$_{12}$, we employed a custom code to fit the vibronic progression with Lorentzian line shapes with the same full width at half maximum. The number of curves used for the fit depends on the number of observed vibronic peaks. We thus calculated the values of the Huang–Rhys factor from the best fits, as described in the Discussion section. Similarly, first-order frequency regions of the UV resonance Raman spectra of C$_8$, C$_{10}$, and C$_{12}$ have been fitted with two Lorentzian curves corresponding to the α and β modes. Together with the analysis of UV-Vis absorption spectra, these data allowed us to compute the values of the nondimensional displacement parameters $\delta_1$ and $\delta_2$ needed for the calculation of the resonance Raman intensities (see the next paragraph).

### Calculation of the resonance Raman intensities
The prediction of resonance Raman intensities, based on Albrecht's theory of resonance Raman, has been extensively explained in the Supplementary Discussion. Our Matlab code calculates the Franck–Condon integrals needed in Albrecht's theory, adopting the displaced harmonic oscillator representation of the vibrational levels of the ground and excited electronic states (see Eq. (27) in the Supplementary Discussion). The evaluation of the Franck–Condon integrals requires the knowledge of the displacement parameters that were found with the joint analysis of UV-Vis absorption spectra and UV resonance Raman spectra (see the previous paragraph). Moreover, we neglect the change of curvature between the ground and excited potential energy surfaces of the α mode (see Discussion). The Franck–Condon integrals have been numerically evaluated by using the formulas reported in ref. 44 (see Supplementary Discussion). Once the Franck–Condon integrals were computed, we made the ratio of the intensities (see Eq. (26) in the Supplementary Discussion) to produce the values reported in Fig. 5.

## Data availability
The UV-Vis absorption and UV resonance Raman spectroscopy data generated in this study have been deposited in the Zenodo database under accession code https://doi.org/10.5281/zenodo.6798673.

## Code availability
The codes that support the findings of this study are available from the corresponding author upon request.

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

## Acknowledgements

P.M., M.T., C.C., P.S., S.P., A.L.B., V.R., and C.S.C. acknowledge funding from the European Research Council (ERC) under the European Union's Horizon 2020 research and innovation program ERC Consolidator Grant (ERC CoG2016 EspLORE grant agreement no. 724610, website: www.esplore.polimi.it). We acknowledge Elettra Sincrotrone Trieste for providing access to its synchrotron radiation facilities and for financial support (proposal number 20205267). We thank Dr. A. Gessini of the IUVS beamline at Elettra for the technical support.

## Author contributions

P.M., S.P., A.L.B., V.R., and C.S.C. conceived the experiment. P.M. and S.P. synthesized the H-capped polyyne samples. P.M., B.R., M.T., V.R., and C.S.C. carried out the UV resonance Raman measurements. M.T., C.C., and P.S. performed theoretical analysis. P.M., M.T., and C.C. performed data analysis. P.M. wrote the paper. All authors discussed the results and contributed to writing subsequent manuscript drafts.

## Competing interests

The authors declare no competing interests.
