## [Peer Review File · Nature Communications]

REVIEWER COMMENTS

Reviewer #1 (Remarks to the Author):

Size-selected polyynes were investigated by UV resonance Raman spectroscopy, allowing the authors to precisely determine the electronic and vibrational structures of the ground and excited states. Interestingly the intensities of the overtones of the C-modes are more pronounced when excited with the lasers of energy close to $|0\rangle_g \rightarrow |1 \text{ or } 2\rangle_e$, which can be well explained by the Albrecht's theory. The study has been systemically performed and the manuscript is very well written. I suggest considering the following comments before acceptance.

1. What were the form of the samples for the Raman measurement? In solution or in powder?
2. Different from the other C10 and C12 polyynes, why 4α of C8 polyyne has a higher relative intensity than the 3α ? Why the 2α of C12 was more intense when excited with $|0\rangle_g \rightarrow |0\rangle_e$, but not $|0\rangle_g \rightarrow |1\rangle_e$ like the other polyynes?
3. I wonder how the δ_1 and δ_2 change with the lengths of the polyynes longer than the C12?

Reviewer #2 (Remarks to the Author):

The paper by Marabotti and co-workers reports an interesting piece of science dealing with electronic and vibrational properties of short linear carbon chains. They investigated by means of theory and experiments domains that were not assessed so far, which are the excitation of the spectra at high energies (UV photons) from brilliant source, so they were able to probe higher order overtones, such as 5th orders, and therefore, calculate a couple of parameters within the theoretical framework employed. The paper is of high quality and certainly merits publication.

However, the authors could improve the paper by correlating in more details how the properties of the short carbyne connects to long ones, such as those reported encapsulated by carbon nanotubes. How are the expectations for other chains with an intermediate number of atoms, such 24, or close

to that size? Are they hard to prepare? Since the chains are small molecules, the authors could comment on the effect of solvent on the results they got?

One minor point is that some recent reference which contributes to advancing the understanding of electronic and vibration properties of long linear carbon chains inside carbon nanotubes were not quoted. Some examples are Carbon 196, 20-28 (2022), Carbon 133, 446-456 (2018), Carbon 90, 172-180 (2015).

In summary, the paper brings novelties to the field and merit publication in Nature Communications.

Dear Reviewers,

Please find below our point-by-point answers to your comments with the indication of the changes made to the manuscript. We reported in red the changes made in the manuscript and in blue the answer to the referees.

Sincerely Yours,

Carlo Casari

Reviewer #1 (Remarks to the Author):

Size-selected polyynes were investigated by UV resonance Raman spectroscopy, allowing the authors to precisely determine the electronic and vibrational structures of the ground and excited states. Interestingly the intensities of the overtones of the C-modes are more pronounced when excited with the lasers of energy close to $|0\rangle_g \rightarrow |1 \text{ or } 2\rangle_e$, which can be well explained by the Albrecht's theory. The study has been systemically performed and the manuscript is very well written. I suggest considering the following comments before acceptance.

1. What were the form of the samples for the Raman measurement? In solution or in powder?

The samples are in solutions. We specified it in the "Methods" section on page 18:

Solutions of size-selected ($n = 8, 10, 12$) hydrogen-capped polyynes (HC_nH , from now on C_n) were synthesized by pulsed laser ablation in acetonitrile, detected, and separated by high-performance liquid chromatography [...]

2. Different from the other C_{10} and C_{12} polyynes, why 4α of C_8 polyyne has a higher relative intensity than the 3α ? Why the 2α of C_{12} was more intense when excited with $|0\rangle_g \rightarrow |0\rangle_e$, but not $|0\rangle_g \rightarrow |1\rangle_e$ like the other polyynes?

As for the first point, since the model is not accurate enough to capture this different behavior, we have no precise explanation for the observed behavior. We added in the main text on pages 15-16 a comment to guide the reader in appreciating the strength and the weakness of the model in justifying the observed experimental trends.

By comparing experimental and theoretical data of Fig. 5, we observe that the model can capture most of the observed modulation of the relative intensities of the overtones for the different resonance conditions, in particular for the $|0\rangle_g \rightarrow |0\rangle_e$ (C_8 and C_{10}) and the $|0\rangle_g \rightarrow |1\rangle_e$ (C_8 , C_{10} , and C_{12}) vibronic transitions. However, in the case of the $|0\rangle_g \rightarrow |2\rangle_e$ resonance condition, the model does not fully account for the observed different behaviors of the three polyynes, and the predicted trends are fairly similar from C_8 to C_{12} . This is expected because the adopted assumptions can lose their robustness for Raman transitions involving higher k .

Remarkably, in C_8 we observe most of the discrepancy, and this is the molecule for which the one-mode approximation is more critical since there are two Raman modes with appreciable δ values, as can be observed in Figure 4c. We then added the following sentence to the main text on page 14

Based on this observation, we expect that the fine details in the resonance Raman trends are harder to describe with the one-mode approximation in C_8 than in C_{12} and longer chains, whose δ_2 is expected to approach zero.

As for the second point, we notice that there is no experimental data available for C_{12} in resonance conditions with the $|0\rangle_g \rightarrow |0\rangle_e$ vibronic transition. Most likely, the referee was misled in reading the plots in Figure 5. Therefore, we changed the experimental panels of Fig. 5 by making explicit reference to the vibronic transitions and we specifically added a sentence in the corresponding figure caption.

Figure 1 [...] The spectrum corresponding to the absorption maximum of C_{12} (273 nm, $|0\rangle_g \rightarrow |0\rangle_e$) could not be recorded due to limitations in the undulator gap aperture. [...]

3. I wonder how the δ_1 and δ_2 change with the lengths of the polyynes longer than the C_{12} ?

We have included a comment on page 17 about the expected behavior of δ_2 in longer chains. δ_1 is expected to reach a plateau value for the infinite chain, corresponding to the observation of a single Raman peak (α).

For longer chains, one expects that the α mode is dominant since δ_2 becomes negligible already for C_{12} . In such conditions, it can be shown that the dispersion of the α mode as a function of the chain length monotonically approaches the limit of about 1900 cm^{-1} for an isolated chain of infinite length⁴⁷. Furthermore, we expect to observe similar intensity modulations at different resonance conditions as those studied in this work.

Reviewer #2 (Remarks to the Author):

The paper by Marabotti and co-workers reports an interesting piece of science dealing with electronic and vibrational properties of short linear carbon chains. They investigated by means of theory and experiments domains that were not assessed so far, which are the excitation of the spectra at high energies (UV photons) from brilliant source, so they were able to probe higher order overtones, such as 5th orders, and therefore, calculate a couple of parameters within the theoretical framework employed. The paper is of high quality and certainly merits publication.

However, the authors could improve the paper by correlating in more details how the properties of the short carbyne connects to long ones, such as those reported encapsulated by carbon nanotubes. How are the expectations for other chains with an intermediate number of atoms, such 24, or close to that size? Are they hard to prepare? Since the chains are small molecules, the authors could comment on the effect of solvent on the results they got?

We have added a comment just before the conclusions where we discuss the issues raised by the referee. First, we have compared our results with the extrapolation of the vibrational properties of longer chains and the effects of the interaction and encapsulation within nanotubes. For chains with intermediate length, we expect that δ_2 becomes negligible and the vibrational properties of the chains can be more exactly described by the one-mode approximation. For example, for chains of more than 20 atoms the β mode disappears, and the first-order region, as well as the overtones, are dominated by the $m\alpha$ modes alone. However, these chains are very unstable and hard to be synthesized with our method (pulsed laser ablation in liquid), also in solution, and their absorption maxima belong to a range of the UV spectrum (270 – 380 nm) in which is difficult to find finely tunable probes.

The effect of the solvent in our model would be accounted for by a modulation of the numerical values of the δ displacement parameters. To assess the relevance of this effect we should carry out experiments in different solvents, which is however not affordable within our time constraints and difficulties in sample preparation. However, we expect minor effects on the δ parameters, and therefore similar trends in different solvents.

We report here the paragraph we added on page 17:

For longer chains, one expects that the α mode is dominant since δ_2 becomes negligible already for C_{12} . In such conditions, it can be shown that the dispersion of the α mode as a function of the chain length monotonically approaches the limit of about 1900 cm^{-1} for an isolated chain of infinite length⁴⁷. Furthermore, we expect to observe similar intensity modulations at different resonance conditions as those studied in this work. Another effect worth investigating with resonance Raman spectroscopy is the behavior of polyynes in different solvents and/or encapsulated in carbon nanotubes. We may expect a similar qualitative behavior as that discussed here for C_8 , C_{10} , and C_{12} , with a possible modulation of the δ parameter because of the interaction effects with the environment. For instance, such effects have been highlighted by comparing the Raman spectra of hydrogen-capped polyynes in solution with those of the same polyynes encapsulated in carbon nanotubes which shows a remarkable decrease of the position of the α mode by a few tens of wavenumbers⁴⁸. Similar experiments have been performed on long linear carbon chains or confined carbyne in carbon nanotubes showing the effects of the confinement on the Raman bands of sp-carbon chains^{49–54}.

One minor point is that some recent reference which contributes to advancing the understanding of electronic and vibration properties of long linear carbon chains inside carbon nanotubes were not quoted. Some examples are Carbon 196, 20-28 (2022), Carbon 133, 446-456 (2018), Carbon 90, 172-180 (2015).

We included some citations to works on confined carbon chains in the paragraph reported just above to improve the comparison with other sp-based carbon systems.

In summary, the paper brings novelties to the field and merit publication in Nature Communications.

REVIEWERS' COMMENTS

Reviewer #1 (Remarks to the Author):

The authors response to my comments very well. I would suggest accepting the revised manuscript in Nature Communications.